# Unpacking how teacher support enhances learning engagement in physical education: A serial mediation model of academic self-efficacy and task orientation among Chinese college students

Xinjian Xu[1,2]*, Yongneng Tan[3]*, Borhannudin bin Abdullah[2]

1 School of Physical Education, Sanming University, Sanming Fu Jian, China, 2 Faculty of Educational Studies, Universiti Putra Malaysia, Selangor, Malaysia, 3 Library, Qingdao University of Science and Technology, Qingdao, Shandong, China

* xuxjsmxy@163.com (XXJ); tanyongneng1985qd@163.com (TYN)

## Abstract

### Background

Despite the Chinese government's focus on student physical education, low student participation in physical activities persists, necessitating efforts to improve engagement.

### Purpose

This study explores the relationship between teacher support and university students' engagement in physical education, focusing on the mediating roles of academic self-efficacy and sport task orientation,

### Methods

A quantitative analysis was conducted to investigate the mediation effects among 2,625 students from various regions and school types.

### Results

The study demonstrated significant correlations between teacher support, academic self-efficacy, sport task orientation, and PELE. Teacher support was found to positively influence physical education learning engagement (PELE) both directly and through academic self-efficacy and sport task orientation. The model fit indices suggested a good fit,

**Data availability statement:** The datasets generated and analyzed during the current study are available in the Figshare repository, DOI: https://doi.org/10.6084/m9.figshare.27940014.

**Funding:** 2025 Integrated Specialized and Innovation Curriculum, Z2025025K; 2025 Integrated Specialized and Innovation Program, Z2025005Z; 2025 University-Level Model Course for Ideological and Political Education, KCSZSFK2519; Undergraduate Teaching Case Repository of Sanming University.

**Competing interests:** The authors have declared that no competing interests exist.

## Conclusion

The findings highlight the critical roles of teacher support, academic self-efficacy, and sport task orientation in enhancing students' engagement in physical education, offering valuable insights for improving physical education practices.

## Introduction

The Chinese Ministry of Education reported that in 2019, only 23.8% of students aged 6–22 passed the standard physical examinations [1], highlighting a critical deficiency in physical fitness among the youth. This low pass rate raises concerns about students' physical well-being and signals a pressing need to enhance the quality of physical education (PE) to achieve broader educational and health objectives. Research indicates that engagement in PE learning not only concerns students' physical health, but it also significantly impacts their academic performance, mental health, and social adaptability [2]. However, persistent challenges such as unhealthy lifestyles and disparities in sports resources across regions contribute to the generally inadequate levels of student engagement in university PE classes [3]. Despite sustained policy reforms aimed at revamping PE curricula, student engagement in PE remains consistently low, reflecting a gap between policy goals and actual student involvement [4]. This study thus addresses the urgent need to understand the factors influencing PE engagement to inform more effective interventions.

## Literature review

Student engagement is an important topic in student education. Fredricks et al. (2004) identified three types of learning engagement: behavioral engagement (BE), affective engagement (AE), and cognitive engagement (CE). Engagement in PE learning is mainly influenced by the environment [5], individual factors [6], school [7], teachers [8], and other social and cultural factors [9]. In a preliminary survey of 100 college students, the results showed that teacher support received the highest scores, indicating that most students believe teacher support has the greatest impact on their learning behaviors in PE classes.

Teacher support refers to various forms of assistance provided by teachers to meet students' academic, emotional, social, and psychological needs [10]. However, much of the existing research focuses on the influence of teacher support on student engagement at the primary and secondary school levels, with limited exploration of its specific effects in higher education physical education. Self-Determination Theory (SDT) posits that teacher support can be categorized into autonomy support, relatedness support, and competence support [11]. Black & Deci (2000) suggest that autonomy support includes giving students the freedom to make choices, thereby promoting their learning initiative. Patrick & Ryan (2009) argue that relatedness support enhances students' learning motivation and academic achievement by fulfilling their need for belonging. Legault et al. (2006) assert that competence support, by providing challenging tasks, enhances students' sense of competence and learning motivation. In PE class,

students expect support in teaching content, learning tasks, practice methods, and interpersonal relationships [12]. Students' perception of support can alter their self-efficacy, meet their diverse needs for sport tasks, and consequently increase their engagement in PE learning [13]. While existing research highlights the role of self-efficacy in general academic contexts, there is a lack of in-depth exploration regarding its specific role in PE, particularly at the university level.

Social Cognitive Theory (SCT) posits that self-efficacy refers to an individual's confidence in their ability to plan and successfully complete specific tasks [14]. Although the positive relationship between self-efficacy and motivation is well-documented in the literature, its application in PE settings has not been fully examined. Individuals with high self-efficacy are more likely to choose challenging academic tasks and persist in them [15]. Self-efficacy positively predicts behavioral, cognitive, and affective [16]. In academic settings, beliefs about academic self-efficacy influence students' academic and career choices [17]. Students with high academic self-efficacy often exhibit stronger learning motivation and achieve better academic outcomes [18]. Yang (2021) found that academic self-efficacy and academic behavioral engagement mediate the relationship between teacher relatedness support and students' mathematics performance [19]. Therefore, academic self-efficacy can influence students' engagement in PE by affecting their sport task orientation.

Achievement Motivation Theory (AMT) posits that sport task orientation influences how athletes perceive and interpret success and failure in physical activities, thereby affecting their achievement goals [20]. Individuals' goal orientations are classified into mastery orientation (or learning orientation) and performance orientation. Athletes with a mastery orientation, who value personal progress and learning, tend to exhibit higher intrinsic motivation and sustained effort [21]. Sport task orientation also impacts athletes' attitudes towards training and competition [22]. For college students, sport achievement goals are crucial for influencing their PE performance. The relationship between high perceived ability in different sports and self-orientation varies, with research indicating that the relationship between students' task achievement orientation and perceived ability depends on specific tasks. This relationship is based on students' self-perceptions of their abilities for specific tasks [23]. Most studies have focused on athletic populations, with limited investigation into how achievement motivation and task orientation impact non-athlete college students in PE classes.

In summary, although research has highlighted the importance of teacher support, self-efficacy, and task orientation for learning engagement, it remains to be studied whether these factors have a significant impact in university PE courses. Moreover, there is still an insufficient understanding of the mechanisms through which these factors influence engagement. This study seeks to address this research gap by examining how teacher support influences university students' engagement in PE through academic self-efficacy and task orientation, offering new perspectives for teaching practices and policy-making in higher education PE.

## Research objectives, question and hypotheses

### Research objectives

This study aims to assess the differences in PE learning engagement among students with various individual backgrounds and to explore the relationship between teacher support (TS) and PE learning engagement (PELE) among Chinese university students, with a particular focus on the mediating roles of academic self-efficacy (ASE) and sport task orientation (STO).

### Research question

Does ASE mediate the relationship between TS and PELE? Does STO mediate the relationship between TS and PELE? Whether there is a serial mediating role between them?

### Research hypotheses

$H_1$: ASE positively mediates the relationship between TS and PELE. $H_2$: STO positively mediates the relationship between TS and PELE. $H_3$: TS influences PE learning engagement through the mediating roles of ASE and STO.

## Methodology

The study employed quantitative analysis methods, using Mplus to conduct confirmatory factor analysis on the pilot sample, and R to verify the scale's omega reliability. Stata was used for data cleaning and structural equation modeling analysis of the sample data to assess the relationships between variables and uncover the underlying mechanisms of influence.

### Population

This study used stratified sampling based on economic regions, school types, and student grades (Table 1).

The first layer is stratified by different regional economic levels, with plans to survey 16 provinces. The second layer is stratified by administrative level and nature, planning to survey 57 schools. The third layer is stratified by student grade, with the population sample focusing only on freshman and sophomore non-PE major university students.

### Sample Size

Based on G*power calculations, the formula for the number of groups is geography * type of college * grade level, resulting in a total of 16*4*2 = 128 groups. The confidence level was chosen to be 95%, because of the large individual differences in students; the effect size was chosen to be a small effect (estimate Cohen's d = 0.2); and the efficacy was set to 0.95, which means that there is a 95% probability of detecting an effect that is actually present. The results were: The total sample size is 1664. Based on the expected response rate of about 60%, about 3000 questionnaires need to be distributed. Due to the large number of provinces in the central, western, and eastern regions, sampling was conducted in the four economic zones at a ratio of 1:3:4:4 to improve measurement accuracy. Actually, a total of 3,575 questionnaires were distributed and all were returned. After data cleaning using Stata, the remaining valid sample consisted of 2,625 responses, with a response rate of 73.4%, involving 57 schools, effectively completing the sample selection process.

### Instruments

The research, targeting Chinese university students, employs scales tailored to the Chinese language environment, instrument selection and psychometric properties:

The Teacher Support Scale used in this study was crafted by Chi (2017) and incorporates elements from the LCQ, Wellborn et al. (1988), and Stornes, Bru, and Idsoe (2008). This scale divides teacher support into three dimensions: autonomy, relatedness, and competence support [24–26]. The psychometric properties of the scale include high internal consistency and construct validity as previously tested in similar educational contexts. Example item: "My teachers try to understand how we see things before giving advice."

**Table 1. Stratified sampling information table for population distribution.**

| economic area | Covered province | Uni. | S.P. | S.A. | Grade |
|---|---|---|---|---|---|
| Northeast | 3: Liaoning, Jilin and Heilongjiang | 258 | 66% | 2 | Freshman And Sophomore |
| East | 10: Beijing, Tianjin, Hebei, Shanghai, Jiangsu, Zhejiang, Fujian, Shandong, Guangdong, Hainan | 1050 | 50% | 3 | |
| Midland | 6: Shanxi, Anhui, Jiangxi, Henan, Hubei and Hunan | 749 | 50% | 5 | |
| West | 12: Inner Mongolia Autonomous Region, Guangxi Zhuang Autonomous Region, Chongqing, Sichuan, Guizhou, Yunnan, Tibet Autonomous Region, Shanxi, Gansu, Qinghai, Ningxia Hui Autonomous Region, Xinjiang Uygur Autonomous Region | 763 | 50% | 6 | |
| Total | 31 Provinces | 2820 | 50% | 16 | 2 |

* By economic region, excluding adult colleges and universities, Hong Kong, Macao and Taiwan, and military colleges and universities; Uni (N): Number of University; S.P.: province sampling probability; S.A.: province sampling amount;

The Academic Self-Efficacy Scale is adapted from Greene et al. (2004), based on the Motivated Strategies for Learning Questionnaire (MSLQ) and the Patterns of Adaptive Learning Scales (PALS) [27]. Reliability and validity indicators for this scale have been established in previous studies to measure students' confidence in their learning abilities accurately. Example item: "I am sure about my ability to do the training tasks after the P.E. class"

Task Orientation is measured using the 8-item "Sport Goal Orientation Scale" developed by Duda and Nicholls [28], a core construct in achievement goal theory, originally proposed for education by Nicholls (1984) and later applied to sports by Duda (1993). This instrument was chosen for its validated use in assessing motivational orientations in sports settings. Example item: "I am focused on mastering skills rather than just winning."

Following Lam et al. (2014) and Skinner et al. (2008), the Engagement Scale used in this study captures engagement in three dimensions: behavioral, affective, and cognitive [29,30]. Psychometric testing has shown strong validity and reliability in assessing student engagement levels. Example item: "I pay attention and participate actively during physical education classes."

This study utilized internationally validated scales, and need to be translated into Chinese. The translation process involved initial translation using DeepL, back-translation by two senior English teachers, and expert review for accuracy. The revised questionnaire was tested with students at Sanming University, followed by further adjustments based on feedback and final refinements during an internal academic seminar.

### Pilot study and data collection

In a pilot study, 500 questionnaires were distributed, yielding 455 valid. Result showed high perceived teacher support, with skewness values near 0, indicating a nearly normal distribution. Confirmatory factor analysis (CFA) using Mplus found that 6 out of 45 standardized loadings were below 0.7, specifically for items AS4, AE5, BE4, BE5, STO7, and STO8.

### Revised CFA model fit indices

Upon revising the model, another CFA test was performed using Mplus. The chi-square value was more than three times the degrees of freedom, and the RMSEA value stood at 0.072, which is within the acceptable limit of less than 0.08. The CFI and TLI values were 0.918 and 0.911, respectively, both surpassing 0.9, indicating strong internal consistency. The SRMR value was 0.048, below the 0.05 threshold, demonstrating an excellent model fit.

### Instrument validity

**Evaluation of convergent validity.** Table 2 shows that most standardized factor loadings for the dimensions were above 0.8, indicating that each indicator was statistically significant. Composite reliability values exceeded 0.8, confirming the reliability of the constructed factors. The average variance extracted (AVE) values were above the 0.5 threshold, demonstrating that the factors accounted for the majority of the variance in the variables, thus indicating excellent convergent validity. Overall, there were strong correlations between each constructed factor and its indicators, the indicators were statistically significant, and the model demonstrated strong reliability.

**Evaluation of discriminant validity.** Using Fornell and Larcker's method for assessing discriminant validity, the square root of the AVE for each latent variable was compared to the correlations among the variables. If the square root of the AVE is greater than the correlation coefficients, the construct is considered to have discriminant validity.

Table 3 demonstrates that the square roots of the AVE values for each dimension were higher than the correlation coefficients between the factors in both horizontal and vertical dimensions, indicating strong discriminant validity.

**Instrument reliability.** We used the Omega (ω) coefficient to assess the internal consistency and factor structure of the scales. The Cronbach's alpha coefficient often imposes a stringent and seldom-met condition in practical applications due to its reliance on the Tau-equivalent assumption [31], making its reporting more theoretical than practical. Consequently, the Omega coefficient is regarded as a more robust and reliable measure of internal consistency [32].

**Table 2. The revised model convergence validity.**

| Factor | TS (AS,CS,RS) | | | PELE (AE,BE,CE) | | | STO | ASE |
|---|---|---|---|---|---|---|---|---|
| Ind. | AS | CS | RS | AE | BE | CE | STO | ASE |
| Std. Loading | 0.911 | 0.986 | 0.961 | 0.842 | 0.876 | 0.863 | 0.826 | 0.865 |
| P-Value | 0.000 | 0.000 | 0.000 | 0.000 | 0.000 | 0.000 | 0.000 | 0.000 |
| CR | 0.968 | | | 0.895 | | | 0.929 | 0.944 |
| AVE | 0.909 | | | 0.740 | | | 0.688 | 0.707 |

\* CR (Composite Reliability); AVE (Average Variance Extracted); TS: teacher support; AS: autonomy support; CS: competence support; RS: relatedness support; PELE: PE learning engagement; AE: affective engagement; BE: behavioral engagement; CE: cognitive engagement; STO: sport task orientation; ASE: academic self-efficacy;

**Table 3. Discriminant validity report.**

| Var | TS | PELE | STO | ASE |
|---|---|---|---|---|
| TS | **0.953** | | | |
| PELE | 0.517 | **0.860** | | |
| STO | 0.534 | 0.826 | **0.830** | |
| ASE | 0.430 | 0.767 | 0.795 | **0.841** |

Calculations were performed in R using the omega function from the psych package, with the dataset path and number of factors as inputs. According to standards, $\omega \geq 0.9$ denotes very high internal consistency [33].

The results show that the overall $\omega$ coefficient is 0.98, the $\omega$ general factor coefficient is 0.80, and the $\omega$ hierarchical coefficient is 0.81. The Schmid-Leiman factor loading matrix indicates the loadings of the general factor and group factors on each variable. For example, the loading of variable AS1 on the general factor is 0.40, while its loading on group factor F1\* is 0.45, with a common factor variance of 0.36 and unique variance of 0.64. The proportion of variance explained by the general factor is 0.44. These results demonstrate that the scale has very high overall reliability and validity, with all factors collectively explaining 98% of the variance in the scale, and the general factor accounting for 80% of the variance. This further confirms the reliability and validity of the scale in measuring the target constructs.

**Ethics and data quality.** This study was approved by the Research Ethics Committee for Human Subjects of Universiti Putra Malaysia (JKEUPM) (Reference No. JKEUPM-2024–497) on October 16, 2024. Participant recruitment was conducted between October 20 and December 30, 2024. An anonymous, self-administered online questionnaire was used to minimize potential risks. Participation was entirely voluntary, and respondents were allowed to skip any question or withdraw from the study at any time without penalty, thus ensuring minimal discomfort and maximum privacy.

All participants were adults and provided written informed consent prior to data collection. The consent process emphasized the voluntary nature of participation and clarified the purpose, procedures, and rights of participants. The study posed no more than minimal risk to participants, while the potential benefits—such as informing improvements in physical education practices and enhancing student engagement—were deemed substantial. To ensure data reliability and accuracy, the questionnaire underwent a pilot study and expert validation prior to formal deployment.

## Results

### Preliminary statistical analysis

This study sampled students from various universities, considering key factors such as education level, grade, gender, and major background. Surveys were distributed across 19 provinces, resulting in 3,575 questionnaires. After data cleaning, 2,625 valid samples from 57 schools were retained. The distribution of student samples by school type indicated that

the majority of students came from provincial colleges (1,647, 63%). Ministry-affiliated colleges, local institutions, and private institutions accounted for 329, 236, and 413 students, respectively. Undergraduate students made up the majority (1,966, 75%), while vocational college students accounted for 659. Freshmen (1,628) outnumbered sophomores (997). Female students (1,713) outnumbered male students (912). Humanities and social sciences majors were slightly more represented (1,428) compared to natural sciences majors (1,197). This comprehensive dataset demonstrates a diverse distribution of student samples across various categories, meeting the research requirements for sample breadth and depth.

## Descriptive statistics

This study primarily examines the impact of teacher support (TS) on PE learning engagement (PELE) and analyzes the mediating roles of academic self-efficacy (ASE) and sport task orientation (STO). TS consists of three subdimensions: autonomy support (AS), competence support (CS), and relatedness support (RS). PELE includes affective engagement (AE), behavioral engagement (BE), and cognitive engagement (CE). The scale comprises 45 items. Since the item weights are not uniform, composite factor scores (CFS) were calculated for descriptive statistics.

Table 4 presents the descriptive statistics and normality of the CFS for each variable (N = 2,625). The mean scores center around zero due to standardization, with standard deviations ranging from 0.93 to 0.99, kurtosis values from 2.63 to 3.90, and skewness values from −0.73 to 0.23. The Shapiro-Wilk W test results, all above 0.90, suggest a balanced distribution with minor deviations from normality.

## Difference analysis

Table 5 reports the homogeneity of variance test and t-test results for group mean differences. The impact of educational level on these variables wasn't significant. The results for major fields of study indicate that students in the sciences and humanities performed similarly. Gender analysis showed no significant differences in BE and AE scores, while cognitive engagement significantly differed between freshmen and sophomores. These results suggest minor influences of student background on PELE.

## Correlation analysis

Table 6 presents the correlation coefficients among four variables (PELE_comp, TS_comp, ASE_comp, and STO_comp). The Pearson correlation coefficients show that PELE_comp has moderate to high positive correlations with the

**Table 4. Descriptive statistics and normality of composite factor scores.**

| CSF | Mean/Prop. | SD | Min. | Max. | Median | Kurtosis | Skew. | W |
|---|---|---|---|---|---|---|---|---|
| TS_comp | .00 | .97 | −4.44 | 1.12 | −.27 | 3.84 | −.71 | 0.91861 |
| AS comp | .00 | .98 | −4.22 | 1.17 | −.18 | 3.90 | −.69 | 0.91746 |
| CS_comp | .00 | .96 | −4.31 | 1.02 | −.31 | 3.64 | −.73 | 0.95601 |
| RS_comp | .00 | .97 | −4.24 | 1.08 | −.25 | 3.57 | −.67 | 0.97465 |
| ASE_comp | .00 | .98 | −3.26 | 1.48 | .17 | 2.63 | −.20 | 0.98641 |
| STO_comp | .00 | .98 | −3.95 | 1.54 | .06 | 2.67 | −.22 | 0.95648 |
| PELE_comp | .00 | .93 | −3.61 | 1.71 | −.09 | 2.37 | .19 | 0.96186 |
| AE_comp | .00 | .98 | −3.37 | 1.48 | .16 | 2.23 | .11 | 0.90439 |
| BE_comp | .00 | .96 | −3.60 | 1.74 | .06 | 2.38 | .10 | 0.97810 |
| CE_comp | .00 | .99 | −3.43 | 1.70 | −.09 | 2.49 | .23 | 0.94385 |

* W: Shapiro–Wilk W test; *-comp: variable composite factor scores (CFS)

**Table 5. Differences in PELE across individuals with different backgrounds.**

| V.D. | Homo. | Edu_level (B/A) | | Major (H/S) | | Gender (M/F) | | Grade (F/S) | |
|---|---|---|---|---|---|---|---|---|---|
| | | F-sig. | T-Sig. 2 | F-sig. | T-Sig. 2 | F-sig. | T-Sig. 2 | F-sig. | T-Sig. 2 |
| PELE | A | 0.000 | 0.032 | 0.943 | 0.62 | 0.000 | 0.000 | 0.021 | 0.146 |
| | non-A | | 0.045 | | 0.62 | | 0.000 | | 0.142 |
| AE | A | 0.000 | 0.317 | 0.993 | 0.812 | 0.015 | 0.000 | 0.493 | 0.025 |
| | non-A | | 0.340 | | 0.813 | | 0.000 | | 0.025 |
| BE | A | 0.003 | 0.004 | 0.553 | 0.229 | 0.000 | 0.000 | 0.238 | 0.02 |
| | non-A | | 0.007 | | 0.23 | | 0.000 | | 0.019 |
| CE | A | 0.002 | 0.065 | 0.292 | 0.957 | 0.015 | 0.146 | 0.325 | 0.953 |
| | non-A | | 0.077 | | 0.957 | | 0.148 | | 0.953 |

*VD (Variable Dimension); Homo (Homoscedasticity); A (Assumption); Non-A (Non-assumption); B/A: (Bachelor's/Associate's Degree); H/S: (Humanities/Sciences); M/F: (Male/Female); F/S: (Freshman/ Sophomore);

**Table 6. Correlation coefficient.**

| | PELE | TS | ASE | STO |
|---|---|---|---|---|
| PELE | 1 | 0.57*** | 0.61*** | 0.71*** |
| TS | 0.55*** | 1 | 0.53*** | 0.61*** |
| ASE | 0.62*** | 0.57*** | 1 | 0.75*** |
| STO | 0.74*** | 0.62*** | 0.76*** | 1 |

Lower-triangular cell report Pearson's correlation, upper-triangular cells are spearman's rank correlation. *** p<0.01 ** p<0.05 * p>0.1

other three variables (0.57, 0.61, and 0.71), all significant at the 0.01 level. The Spearman rank correlation coefficients confirm these positive relationships (0.55, 0.62, and 0.74, respectively), which are also significant at the 0.01 level. Notably, STO and PELE displayed the strongest relationship, indicating STO's role in enhancing engagement.

## Structural equation model analysis

The structural equation model (SEM) in Fig 1 examined the impact of TS on PELE and the mediating effects of ASE and STO. The analysis included 45 observed variables. TS includes AS, CS, and RS as dimensions; PELE includes AE, BE, and CE dimensions. This SEM analysis, processed with Stata, yielded path coefficients that highlight ASE and STO's mediating effects.

Table 7 presents the model fitting results for a sample size of 2,625. The likelihood ratio chi-square test comparing the model to the saturated model ($\chi^2(30) = 250.636$, $p<0.001$) and the baseline to the saturated model ($\chi^2(45) = 19292.986$, $p<0.001$) indicate significant differences. The RMSEA is 0.075, with a 90% confidence interval of 0.067 to 0.083 and a p-value close to 0.000. The CFI and TLI are 0.988 and 0.979, respectively, both exceeding the recommended threshold of 0.95. The SRMR is 0.018, well below the threshold of 0.08, indicating a good fit.

## The mediating role of academic self-efficacy

The results regarding the mediating role of ASE indicate that the model has a good fit (Fig 2). TS significantly positively affects ASE (path coefficient=0.59, $p<0.001$), and ASE significantly positively influences PELE (path coefficient=0.49, $p<0.001$). Additionally, TS also directly impacts PELE (path coefficient=0.31, $p<0.001$).

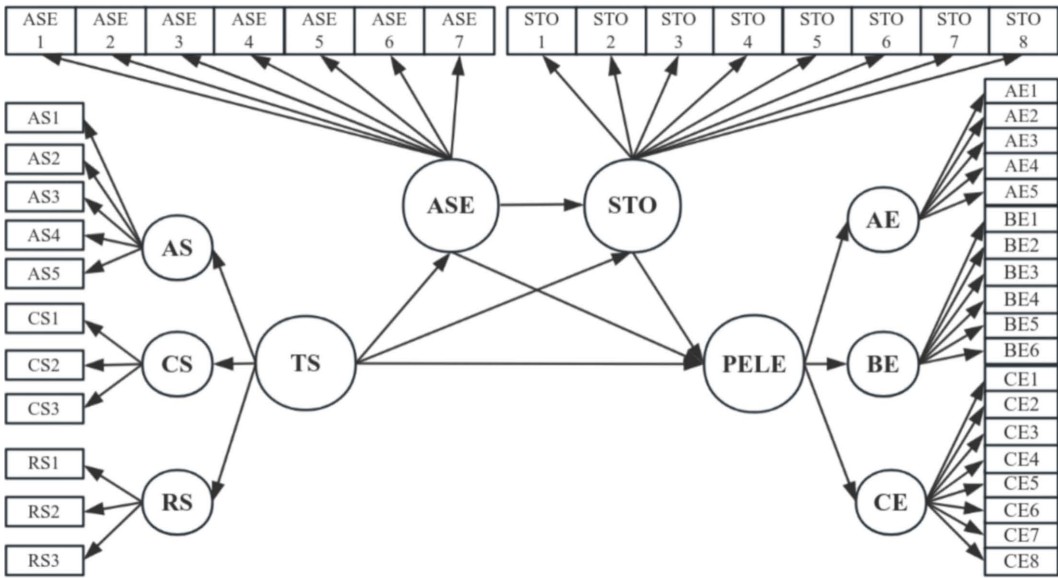

**Fig 1. Structural equation model.**

## The mediating role of sport task orientation

The results regarding the mediating role of STO indicate that the model has a good fit (Fig 3). TS significantly positively affects STO (path coefficient = 0.63, p < 0.001), and STO significantly positively influences PELE (path coefficient = 0.68, p < 0.001). Additionally, TS also directly impacts PELE (path coefficient = 0.17, p < 0.001).

## The serial mediating role

The results of the serial mediating model, as presented in Fig 4 and Table 8, demonstrate a significant positive influence of TS on both ASE and STO. Specifically, the path coefficients are 0.58 for TS to ASE (p < 0.001) and 0.30 for TS to STO (p < 0.001). ASE, in turn, has a significant positive effect on STO, with a path coefficient of 0.59 (p < 0.001). Both ASE and STO significantly positively impact PELE, with path coefficients of 0.12 (p < 0.001) and 0.60 (p < 0.001), respectively. Additionally, TS directly affects PELE, with a path coefficient of 0.15 (p < 0.001). The sub-dimensions analysis reveals that TS significantly influences AS, CS, and RS with path coefficients of 0.90 (p < 0.001), 0.95 (p < 0.001), and 0.95 (p < 0.001), respectively. Similarly, the sub-dimensions of PELE—AE, BE, and CE—have path coefficients of 0.89 (p < 0.001), 0.86 (p < 0.001), and 0.83 (p < 0.001), respectively.

## Discussion

### Interpretation of findings

The study highlights the crucial role of teacher support in enhancing PE learning engagement among Chinese university students. The Structural Equation Model analysis suggests an acceptable model fit, despite the model's limitations given the sample size. This model includes two mediator sub-models and a serial mediating model, which provide insights into the dynamics of teacher support, academic self-efficacy, sport task orientation, and physical education learning engagement.

Specifically, TS positively impacts ASE and STO, which in turn positively affect PELE. This suggests that teacher support first strengthens students' academic confidence, and this confidence motivates them to adopt a mastery-focused task orientation. Such task orientation channels self-belief into sustained effort, ultimately leading to greater PELE. This

**Table 7. Model fitting results of 2625 samples.**

| Fit statistic | Value | Description |
|---|---|---|
| Likelihood ratio | | model vs. saturated |
| chi2_ms(30) | 250.636 | baseline vs. saturated |
| p > chi2 | 0.000 | |
| chi2_bs(45) | 19292.986 | |
| p > chi2 | 0.000 | |
| Population error | | Root mean squared error of |
| RMSEA | 0.075 | approximation |
| 90% CI, lower bound | 0.067 | Probability RMSEA<= 0.05 |
| upper bound | 0.083 | |
| pclose | 0.000 | |
| Baseline comparison | | Comparative fit index |
| CFI | 0.988 | Tucker–Lewis index |
| TLI | 0.979 | |
| Size of residuals | | |
| SRMR | 0.018 | Standardized root mean squared residual |

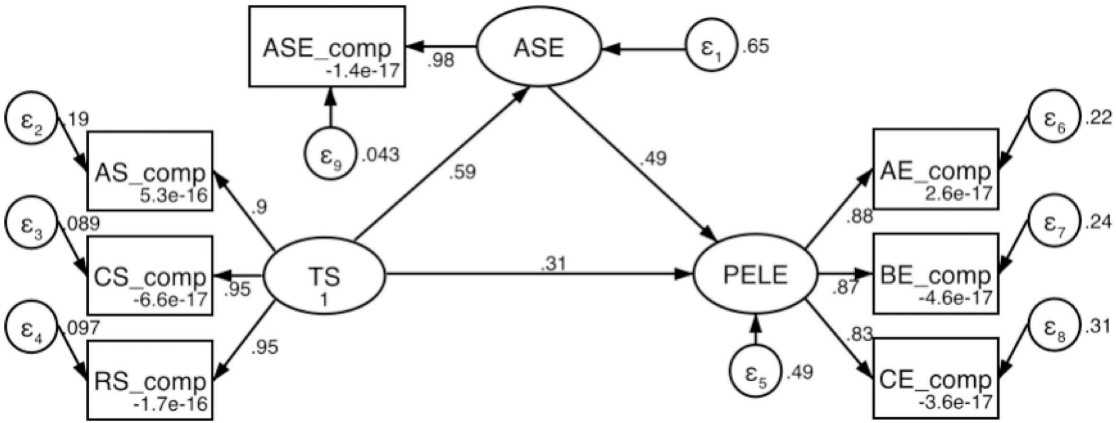

**Fig 2. The mediating model of academic self-efficacy.**

underscores the importance of fostering a supportive educational environment that enhances students' self-efficacy and task orientation in sports to improve their engagement in university PE activities.

The analysis of direct effects shows that TS has a significant positive impact on PELE, underscoring the crucial role of teacher support in motivating and engaging students in PE activities. This finding also corroborates previous research in this field, which demonstrates that teacher support for student autonomy promotes the fulfillment of basic psychological needs, intrinsic motivation [34], positive emotional experiences [35], academic achievement and persistence [36]. This direct effect highlights the importance of teacher encouragement to enhance student engagement in PE.

The analysis of mediating roles shows that the significant mediating roles of ASE and STO in the relationship between TS and PELE. TS positively influences ASE and STO. In the serial mediation model, ASE significantly affects STO. Both ASE and STO significantly impact PELE, respectively. The findings of this study shows that ASE partially mediates the relationship between TS and PELE, underscoring the importance of enhancing students' academic self-efficacy to boost

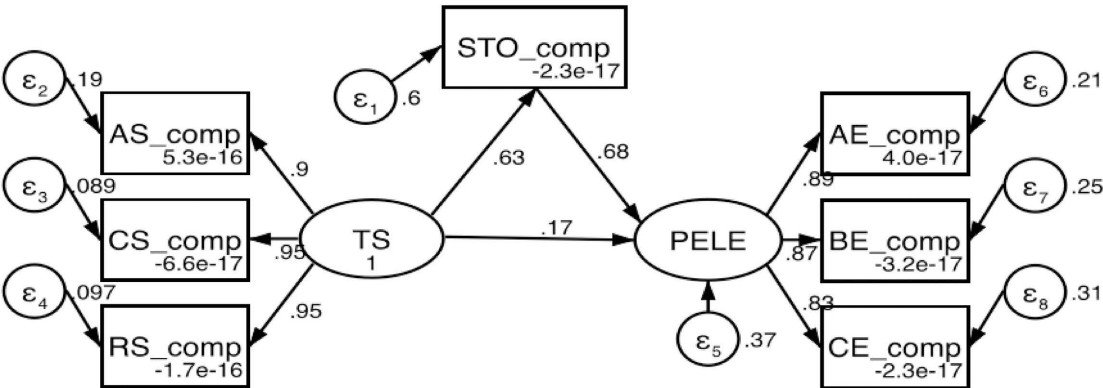

**Fig 3. The mediating model of sport task orientation.**

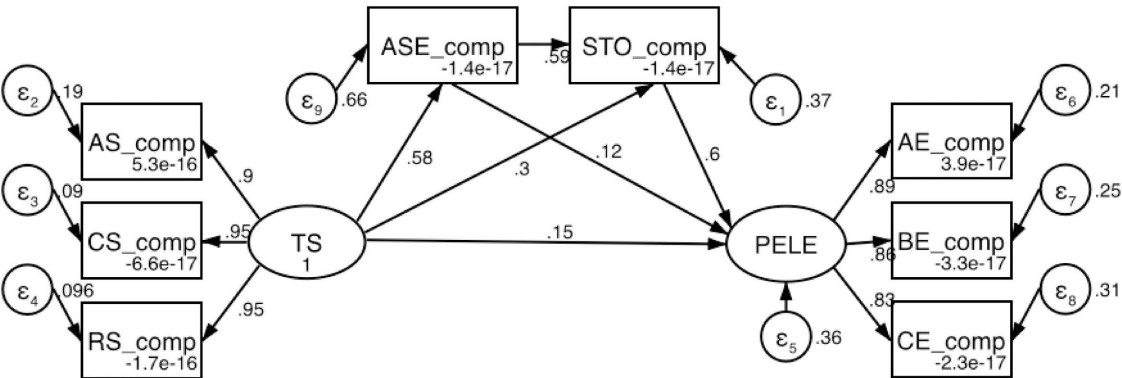

**Fig 4. The serial mediating model.**

their engagement in PE learning, thus validating Hypothesis 1. Similarly, STO partially mediates the relationship between TS and PELE, emphasizing the value of improving students' sport task orientation to foster their PE learning engagement, thereby supporting Hypothesis 2. Moreover, ASE and STO serve as serial mediators in the TS-PELE relationship, highlighting the compounded effect of these factors in increasing student engagement in PE learning, which validates Hypothesis 3.

## Comparison with previous research

**Theoretical background.** In educational psychology, teacher support, self-efficacy, and task orientation constitute essential theoretical frameworks influencing student learning engagement. Teacher support theory, particularly informed by Self-Determination Theory, emphasizes the critical role of teachers in meeting students' basic psychological needs—namely autonomy, competence, and relatedness [37,38]. Simultaneously, Self-efficacy, as posited by Social Cognitive Theory, suggests that students' confidence in their ability to complete learning tasks significantly affects their motivation and behavior [39]. Achievement Goal Theory posits that students' task orientation—their focus on personal learning and mastery—correlates positively with learning engagement [40]. Self-Determination Theory, Social Cognitive Theory, and Achievement Motivation Theory were used to support this study. Together, these perspectives explain the sequential pathway: SDT highlights that teacher support satisfies basic needs and builds ASE; SCT emphasizes that

**Table 8. Observed information matrix.**

| Structural | | Coefficient | | std. err. | z | P > |z| | [95% conf. interval] |
|---|---|---|---|---|---|---|---|
| ASE_comp | TS | .5795917 | .0133866 | 43.3 | 0.000 | .5533544 | .6058291 |
| STO_comp | ASE_comp | .5858884 | .0133396 | 43.92 | 0.000 | .5597433 | .6120336 |
| | TS | .295283 | .0148832 | 19.84 | 0.000 | .2661125 | .3244535 |
| PELE | STO_comp | .6023301 | .020815 | 28.94 | 0.000 | .5615334 | .6431267 |
| | ASE_comp | .1162261 | .0220965 | 5.26 | 0.000 | .0729176 | .1595345 |
| | TS | .1483391 | .0183328 | 8.09 | 0.000 | .1124074 | .1842708 |
| AS_comp | TS | .9010734 | .0042168 | 213.7 | 0.000 | .8928083 | .9093378 |
| CS_comp | TS | .953756 | .0027192 | 350.7 | 0.000 | .9484262 | .9590854 |
| RS_comp | TS | .9505484 | .0028089 | 338.4 | 0.000 | .945043 | .9560537 |
| AE_comp | PELE | .8911135 | .0057948 | 153.8 | 0.000 | .8797559 | .9024712 |
| BE_comp | PELE | .8636615 | .0064426 | 134.1 | 0.000 | .8510342 | .8762889 |
| CE_comp | PELE | .8316448 | .0074353 | 111.8 | 0.000 | .8170718 | .8462178 |

* LR test of model vs. saturated: chi2(30) = 389.16.; Prob > chi2 = 0.0000; *-comp: variable composite factor scores (CFS)

ASE enhances students' motivation to persist; AMT clarifies how STO transforms this confidence into mastery-driven engagement.

**Comparison of research findings.** In comparison to prior research, this study contributes significantly by emphasizing the interplay between TS, ASE, and STO in PELE.

Specifically, this study finds that teacher support has a significant positive impact on students' ASE, corroborating findings by Cheon et al. (2012) and Ryan & Grolnick (1986), which underscore the importance of teacher support for students' autonomy and psychological adjustment. Moreover, it confirms the pivotal role of emotional support from teachers in enhancing students' academic self-efficacy [41–43], providing empirical backing to these relationships.

Previous studies [16,18] have indicated that academic self-efficacy influences learning motivation and strategies. This study expands upon this by clarifying the partial mediating role of ASE between teacher support and PE learning engagement through Structural Equation Modeling, thus offering new empirical evidence.

This research validates the mediating role of task orientation in the relationship between teacher support and PE learning engagement, supporting the findings of Chen and Xiang regarding the significance of task orientation in physical education [13,44]. Notably, in contemporary student-centered teaching models, task orientation is shown to enhance students' learning engagement [45].

**Critical analysis.** Despite these insights, several aspects warrant further exploration. The relatively small sample size in this study may limit the generalizability of the findings. Future research should aim to employ larger, more diverse samples utilizing both cross-sectional and longitudinal designs to investigate the long-term effects of teacher support on student learning engagement.

Most prior studies have focused on Western educational environments, whereas this research is situated within a Chinese context. This presents a new perspective on understanding the role of cultural differences in learning engagement. In Chinese universities, PE is often exam-oriented and teacher-centered, which means students rely heavily on teacher encouragement to build ASE. Once this confidence is established, it becomes a key driver for adopting STO, as students gradually shift from compliance to active mastery. Future studies could further explore how teacher support functions across various cultural settings.

While this study highlights the sequential mediating effects of ASE and STO, it does not delve into other potentially influential factors such as peer support and family environment. Future research should examine how these factors interact with teacher support to affect student learning engagement.

In summary, this study demonstrates innovation and progress in theoretical application, comparison of research findings, and critical analysis, thereby providing new insights and empirical support for understanding the roles of teacher support, self-efficacy, and task orientation in enhancing university students' engagement in physical education.

### Theoretical implications

The results support SDT, SCT, and AMT. These findings highlight the importance of a supportive educational environment that fulfills students' autonomy, competence, and relatedness needs, as suggested by SDT. Furthermore, the significant roles of STO and ASE align with SCT's emphasis on self-efficacy and AMT's focus on achievement goals, indicating that enhancing these mediators can effectively boost student engagement.

### Practical implications

This cross-sectional study suggests a significant association between TS and increased student PELE. Educators should consider implementing strategies that enhance students' STO and ASE, such as creating autonomy-supportive environments, setting challenging yet attainable goals, and providing positive feedback. These approaches could make physical education more engaging and motivating for students, promoting better health and academic outcomes.

### Limitations

It's employed online sampling, encompassing a diverse range of regions. However, the prolonged duration of data collection may have resulted in inconsistent questionnaire response quality. Although the study surveyed participants from 16 provinces, it did not cover all regions of China, limiting the generalizability of the findings.

### Future research

Future research should investigate the longitudinal impacts of TS on PELE, focusing on how variations in STO and ASE over time affect long-term engagement. Additionally, examining the effects of different types of teacher support across varied educational contexts and cultural backgrounds could help generalize these findings further.

## Conclusion

This study explored the relationship of TS and PELE among Chinese university students, with a focus on the mediating roles of ASE and STO. The results revealed significant direct and indirect effects of TS on PELE. Teacher support enhances both ASE and STO, which in turn positively influence students' engagement in physical education activities. Importantly, ASE further reinforces STO, highlighting a compounded positive effect on PELE. Additionally, the model fit indices, including RMSEA, CFI, TLI, and SRMR, confirmed a good fit, validating the proposed model.

These findings align with and extend previous research by demonstrating that teacher support strengthens self-efficacy and task orientation, thereby deepening students' engagement in physical education. This underscores the importance of fostering supportive educational environments that enhance students' self-efficacy and task orientation in sports, ultimately improving their engagement in PE.

The study contributes to the literature by illuminating specific pathways through which teacher support fosters higher engagement in physical education. These findings provide actionable insights for educators and policymakers, suggesting that fostering supportive environments can enhance students' health and academic outcomes. Future research should explore the longitudinal effects of teacher support on PE engagement and examine the influence across different educational and cultural contexts to further substantiate these conclusions.

## Acknowledgments

The authors would like to thank Dr. H.M. from Rice University for his valuable guidance on data analysis, as well as the Department of Sports Training at Universiti Putra Malaysia (UPM) and the School of Physical Education and Health at Sanming University for their support.

## Author contributions

**Conceptualization:** xinjian xu, Yongneng Tan.

**Investigation:** xinjian xu, Yongneng Tan.

**Methodology:** xinjian xu, Borhannudin bin Abdullah.

**Supervision:** Borhannudin bin Abdullah.

**Writing – original draft:** xinjian xu.

**Writing – review & editing:** xinjian xu, Yongneng Tan, Borhannudin bin Abdullah.

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
