## [Decision Letter · Decision Letter 0]

5 Sep 2025

Dear Dr.  xu,

Thank you for submitting your manuscript to PLOS ONE. After careful consideration, we feel that it has merit but does not fully meet PLOS ONE’s publication criteria as it currently stands. Therefore, we invite you to submit a revised version of the manuscript that addresses the points raised during the review process.

We look forward to receiving your revised manuscript.

Kind regards,

Rogis Baker, Ph.D

Academic Editor

PLOS ONE

Journal Requirements:

“2025 Integrated Specialized and Innovation Curriculum, Z2025025K; 2025 Integrated Specialized and Innovation Program, Z2025005Z; 2025 University-Level Model Course for Ideological and Political Education, KCSZSFK2519; Undergraduate Teaching Case Repository of Sanming University”

Reviewer's Responses to Questions

**Comments to the Author**

1. Is the manuscript technically sound, and do the data support the conclusions?

Reviewer #1: Yes

Reviewer #2: Yes

2. Has the statistical analysis been performed appropriately and rigorously?

Reviewer #1: Yes

Reviewer #2: Yes

3. Have the authors made all data underlying the findings in their manuscript fully available?

Reviewer #1: Yes

Reviewer #2: Yes

4. Is the manuscript presented in an intelligible fashion and written in standard English?

Reviewer #1: Yes

Reviewer #2: Yes

Reviewer #1: No comments to the authors. I am not a quantitative researcher, but it appears that your quantitative analysis is well done. The data you provide are sufficient. See responses to questions above. I have no concerns about research ethics or publication ethics.

Reviewer #2: In the paper, the theoretical discussion on the mechanism of action is characterized by "being more descriptive than explanatory". It can illustrate the "composition of the mechanism", but fails to deeply unpack the core logic of the serial mediation mechanism and does not fully answer the question of "why the mechanism exists". It is suggested to add in-depth exploration of the model mechanism and analysis of the local context to enhance theoretical explanatory power. However, this is a personal suggestion; if the focus of this paper is only on describing the existence of the mechanism rather than conducting in-depth theoretical analysis, no revision is necessary.

**Do you want your identity to be public for this peer review?** For information about this choice, including consent withdrawal, please see our Privacy Policy

Reviewer #1: **Yes: ** Linda Billings

Reviewer #2: No

---

## [Author Response · Author response to Decision Letter 1]

9 Sep 2025

Rebuttal letter

Manuscript ID: [PONE-D-25-38320] Title: [Unpacking how teacher support enhances learning engagement in physical education: a serial mediation model of academic self-efficacy and task orientation among Chinese college students]

Dear Editor,

We thank you and the reviewers for the insightful comments and constructive suggestions, which have greatly helped us improve our manuscript. We have carefully considered each point and made the corresponding revisions in the manuscript. Below, we provide a point-by-point response to each comment. The specific changes are also available in the “Revised manuscript (tracked changes copy)” file.

First part, Response Journal Requirements:

Response: We sincerely thank the Editor for reminding us to carefully check the manuscript against the PLOS ONE style requirements. We have revised the manuscript text, tables, and figures according to the formatting guidelines, and ensured compliance with the journal’s file naming conventions.

Regarding the in-text citation style, our manuscript currently uses parentheses “( )” instead of brackets “[ ]”. This formatting arises from the citation style settings in our reference management software (EndNote), which cannot be fully converted into bracket style in our local environment. We kindly request the editorial office to assist in converting the parentheses into brackets at the production stage. If required, we are willing to provide the complete reference library file (EndNote) to facilitate this process.

We hope this explanation is acceptable, and we confirm that apart from this point, all other aspects of the manuscript formatting now conform to PLOS ONE requirements.

2/3. We note that the grant information you provided in the ‘Funding Information’ and ‘Financial Disclosure’ sections do not match. When you resubmit, please ensure that you provide the correct grant numbers for the awards you received for your study in the ‘Funding Information’ section. Please include this amended Role of Funder statement in your cover letter; we will change the online submission form on your behalf.

Response: We thank the Editor for pointing out the inconsistency between the Funding Information and Financial Disclosure sections. We have carefully revised these sections to ensure that the grant numbers and funding details are now consistent. In addition, we have removed the conflicting statement from the manuscript and clarified the corrected funding information in the cover letter.

Response: We would like to clarify the Data Availability Statement. The datasets are already publicly accessible in the Figshare repository (DOI: 10.6084/m9.figshare.30084220), as stated in the manuscript. The earlier discrepancy arose because we mistakenly selected the “available upon acceptance” option in the submission system. This has been corrected in the resubmission, and the information is now consistent across both the manuscript and the submission form.

Response: We confirm that the ethics statement has been retained only in the Methods section. Any duplicate statements appearing elsewhere in the manuscript have been removed to ensure consistency with the journal’s requirements.

Response: Neither Reviewer #1 nor Reviewer #2 requested citation of any specific published works. We have carefully re-checked their comments and confirmed that no additional references were suggested. Therefore, no changes were made to the reference list in this regard.

Response: We have carefully reviewed our entire reference list and confirm that none of the cited papers have been retracted. Therefore, no changes were required in this regard.

The second part: Response Reviewer's Questions

Response to Reviewer 1:

We sincerely thank Reviewer #1 for the positive evaluation of our study, including the acknowledgment that the quantitative analysis is well conducted and the data provided are sufficient.

Response to Reviewer 2:

We greatly appreciate Reviewer #2’s thoughtful comments regarding the theoretical discussion. We agree that the serial mediation mechanism could benefit from deeper theoretical unpacking. Our study was primarily designed as a cross-sectional correlational investigation. Given the broad scope and large, diverse sample, it was not feasible to implement causal comparisons or longitudinal control. Therefore, the focus of this manuscript is to examine the associations among teacher support, academic self-efficacy, sport task orientation, and physical education learning engagement rather than to establish causal effects.

Nevertheless, we found the reviewer’s suggestion very valuable. In response, we have revised the Discussion section to provide stronger theoretical grounding and explanatory depth.

A: added clarifying sentences in the Discussion section to strengthen the explanation of the mechanism and highlight the local educational context;

B: clarified in the rebuttal that the main focus of this paper is to demonstrate the existence and structure of the mechanism rather than to develop an extended theoretical model.

Specifically, we analyzed the mechanism from two theoretical perspectives—Self-Determination Theory (SDT) and Social Cognitive Theory (SCT) combined with Achievement Motivation Theory (AMT)—to clarify why teacher support enhances academic self-efficacy and task orientation, which in turn foster learning engagement. These additions strengthen the theoretical basis of our model and enhance the explanatory power of the findings while remaining consistent with the scope of the study.

The specific revision process is as follows

1.line 419-422 (Track Changes): add, This suggests that teacher support first strengthens students’ academic confidence, and this confidence motivates them to adopt a mastery-focused task orientation. Such task orientation channels self-belief into sustained effort, ultimately leading to greater PELE.

Statement: In the Interpretation of findings, add the logical chain explaining why ASE → STO → PELE.

2.line 456-460 (Track Changes): Together, these perspectives explain the sequential pathway: SDT highlights that teacher support satisfies basic needs and builds ASE; SCT emphasizes that ASE enhances students’ motivation to persist; AMT clarifies how STO transforms this confidence into mastery-driven engagement.

Statement: In the Theoretical implications, add a linked explanation of the three theories rather than listing them separately.

3.line 487-491 (Track Changes): In Chinese universities, PE is often exam-oriented and teacher-centered, which means students rely heavily on teacher encouragement to build ASE. Once this confidence is established, it becomes a key driver for adopting STO, as students gradually shift from compliance to active mastery.

Statement: In the Critical analysis, add the unique features of the TS → ASE → STO pathway in the Chinese context.

---

## [Editor Report · Decision Letter 1]

18 Sep 2025

Unpacking how teacher support enhances learning engagement in physical education: a serial mediation model of academic self-efficacy and task orientation among Chinese college students

PONE-D-25-38320R1

Dear Xu,

We’re pleased to inform you that your manuscript has been judged scientifically suitable for publication and will be formally accepted for publication once it meets all outstanding technical requirements.

Kind regards,

Rogis Baker, Ph.D

Academic Editor

PLOS ONE
---

## [Editor Report · Acceptance letter]

PONE-D-25-38320R1

PLOS ONE

Dear Dr. xu,

I'm pleased to inform you that your manuscript has been deemed suitable for publication in PLOS ONE. Congratulations! Your manuscript is now being handed over to our production team.

Kind regards,

on behalf of

Dr. Rogis Baker

Academic Editor

PLOS ONE